Comprehensive evaluation of fluroxypyr herbicide on physiological parameters of spring hybrid millet

Guo Meijun
Shen Jie
Song Xi-e
Dong Shuqi
Wen Yinyuan
Yuan Xiangyang yuanxiangyang200@163.com
Guo Pingyi pyguo126@126.com
Agronomy College, Shanxi Agricultural University , Taigu , China
Tiessen Axel
Electronic publication date: 2019 Sep 26
Publication date: 2019
Volume: 7
Electronic Location ID: e7794
Received 2019 Mar 18; Accepted 2019 Aug 29
Copyright: © 2019 Guo et al.
Copyright year: 2019
Copyright holder: Guo et al.
License: This is an open access article distributed under the terms of the Creative Commons Attribution License, which permits unrestricted use, distribution, reproduction and adaptation in any medium and for any purpose provided that it is properly attributed. For attribution, the original author(s), title, publication source (PeerJ) and either DOI or URL of the article must be cited.
License URL: https://creativecommons.org/licenses/by/4.0/

Keywords: Photosynthetic characteristics, Comprehensive evaluation, Peroxidation characteristics, Fluroxypyr, Endogenous hormone, Foxtail millet

Funding: National Modern Agriculture Technology CARS-06-13.5-A28 Scientific and Technological Project in Shanxi Province, China 20150311016-2 Key Research and Development General Project in Shanxi Province, China 201603D221003-2 Key Scientific and Technological Project of Shanxi Province 2015-TN-09 Top Young Innovative Talents of Shanxi Agricultural University TYIT201406 This work was supported by the system of National Modern Agriculture Technology (CARS-06-13.5-A28), the Scientific and Technological Project in Shanxi Province, China (20150311016-2), the Key Research and Development General Project in Shanxi Province, China (201603D221003-2), the Key Scientific and Technological Project of Shanxi Province (2015-TN-09), and the Program for the Top Young Innovative Talents of Shanxi Agricultural University (TYIT201406). The funders had no role in study design, data collection and analysis, decision to publish, or preparation of the manuscript.

==============================
Foxtail millet (Setaria italic L.) is an important food and fodder crop that is cultivated worldwide. Quantifying the effects of herbicides on foxtail millet is critical for safe herbicide application. In this study, we analyzed the effects of different fluroxypyr dosages on the growth parameters and physiological parametric of foxtail millet, that is, peroxidation characteristics, photosynthetic characteristics, and endogenous hormone production, by using multivariate statistical analysis. Indicators were screened via Fisher discriminant analysis, and the growth parameters, peroxidation characteristics, photosynthesis characteristics and endogenous hormones of foxtail millet at different fluroxypyr dosages were comprehensively evaluated by principal component analysis. On the basis of the results of principal component analysis, the cumulative contribution rate of the first two principal component factors was 93.72%. The first principal component, which explained 59.23% of total variance, was selected to represent the photosynthetic characteristics and endogenous hormones of foxtail millet. The second principal component, which explained 34.49% of total variance, represented the growth parameters of foxtail millet. According to the principal component analysis, the indexes were simplified into comprehensive index Z, and the mathematical model of comprehensive index Z was set as F = 0.592Z1 + 0.345Z2. The results showed that the comprehensive evaluation score of fluroxypyr at moderate concentrations was higher than at high concentrations. Consequently, one L (active ingredient, ai) ha−1 fluroxypyr exerted minimal effects on growth parameters, oxidase activity, photosynthetic activity, and endogenous hormones, and had highest value of comprehensive evaluation, which had efficient and safe benefits in foxtail millet field.

Introduction

Foxtail millet (Setaria italica L.) is a valuable economic crop because it is rich in protein and crude fat and used as a staple food worldwide. This plant has seven essential amino acids, and its contents are higher than those found in other crops. In particular, methionine and tryptophan, which are important in preventing atherosclerosis and softening the blood vessel, are abundant in this plant. However, weed infestation has severely limited foxtail millet production in China (Guo et al., 2018). Weeds are the most important biotic factor affecting agricultural production; they are responsible for over 55.56% of total foxtail millet yield losses (Zhou et al., 2012). For weed control, herbicide application is considered the most cost-efficient and effective method (Rubin, 1996). Rational application of herbicides, such as monosulfuron, monosulfuron mix propazine, 2,4-D, and prometryn, can effectively control weeds in foxtail millet fields; however, these herbicides readily cause phytotoxicity reaction (Tian & Wang, 2010). Thus, finding suitable herbicides for use in foxtail millet field is a considerable challenge in weed control.

Fluroxypyr (4-amino-3, 5-dichloro-6-fluoro-2-pyridyloxyacetic acid) is originally applied in cereal, olive tree, and fallow cropland fields, to control annual or perennial weeds (Hellou et al., 2009). These herbicides cause auxin overdose or excessive endogenous auxin concentrations, thereby resulting in an imbalance of auxin homeostasis and interaction with other hormones in tissues, which ultimately cause the succeeding series of biochemical and physiological processes associated with herbicide action (Grossmann, 2010). Liu (2014) demonstrated that the label fluroxypyr dose can be used in maize (Zea mays) and winter wheat (Triticum aestivum) fields, and has desirable control effects on broadleaf weeds. Considering the significant differences in the sensitivity of crop varieties to fluroxypyr, it is necessary to indicate whether fluroxypyr is safe for use on foxtail millet.

Foxtail millet tolerance toward herbicides depends on either the alteration of the target site (TS) or non-target site (NTS) mechanisms. TS resistance mechanism is manifested by structural changes due to point mutations in herbicide-binding proteins, such as the D1 protein in the photosystem II (PSII) complex (Thiel & Varrelmann, 2014), acetolactate synthase (Tranel, Wright & Heap, 2014), acetyl-CoA carboxylase (Kaundun, 2014), or 5-enolpyruvylshikimate-3-phosphate synthase (EPSPS) (Sammons & Gaines, 2014). TS resistance mechanism is occasionally the result of an increased number of copies of the target gene, such as EPSPS (Gaines et al., 2010; Vila-Aiub et al., 2014). The NTS mechanism involves herbicide detoxification by glutathione S-transferase or cytochrome P450 monooxygenase, reduced absorption or translocation in the plant and sequestration into vacuoles (Matzrafi et al., 2014). In contrast to TS, NTS resistance mechanisms can occasionally be more widespread and exhibit resistance to many herbicides. However, information regarding the physiological mechanism of Zhangzagu hybrid millet after the application of fluroxypyr is limited. Thus, the NTS resistance mechanisms to fluroxypyr in spring hybrid millet should be understood.

The hybrid foxtail millet “Zhangzagu” is popular, especially Zhangza 5, with a large production area in China due to its high yield, drought resistance, and nutritional value (Dong et al., 2014). Thus, we used Zhangza 5 as the model species for the present study, especially for the study of the physiological mechanism due to tolerance against fluroxypyr. Herein, the objective of this study are as follows: (1) to comprehensively investigate the influence of different dosages of fluroxypyr on the growth and physiological mechanism of spring hybrid millet; (2) to further understand the NTS resistance mechanisms to fluroxypyr-induced oxidative stresses; (3) to compare and analyze the main factors contributing to the effect of fluroxypyr on spring hybrid millet growth. Thus, we provided convincing evidence for the correlations of NTS resistance to fluroxypyr exposure, and aimed at obtaining the optimal fluroxypyr application dose for spring hybrid millet.

Materials and Methods

Plant and experimental design

Fluroxypyr (20%, emulsifiable concentrate) was provided by Dow AgroScience Co. (Jiangsu, China). Foxtail millet (S. italica L. cv. Zhangza 5) was supplied by the Zhangjiakou Academy of Agricultural Sciences of Hebei Province, China. The seeds were uniformly sown in a plastic pot (130 mm diameter) containing a mixture of sand and soil (1:2, v:v). Calcareous cinnamon soil, with a pH of 7.85, 24.49 g kg−1 organic matter, 51.92 mg kg−1 total nitrogen, 24.13 mg kg−1 available phosphorus, and 183.6 mg kg−1 rapidly-available potassium was used in this work. At the three-leaf stage, the foxtail millet seedlings were thinned and maintained at 10 uniform plants per pot.

The experiment was conducted as a randomized complete block design with three replications. After growing to the five-leaf stage, foxtail millet seedlings were treated with 0, 0.5, 1, 2, and 4 L ai ha−1 fluroxypyr by using a laboratory pot-sprayer equipped with a nozzle and previously calibrated to deliver 450 L ha−1. The manufacturer recommended an effective dose of one L ai ha−1 for field application. After herbicide treatment for 5 and 10 days, the physiological indices of all foxtail millet seedlings were analyzed.

Measurement of growth parameters

The plant height, length and width of leaves were measured with a ruler. Leaf area was calculated using the following equation: leaf area = 0.75 × leaf length × leaf width. Leaf width was considered the widest part of the penultimate leaf.

Determination of superoxide generation rate and H2O2 content

Fresh foxtail millet leaves (0.1 g) were homogenized with two mL of 65 mmol L−1 sodium phosphate buffer (pH 7.8) and centrifuged at 10,000 rpm for 10 min. The supernatant (one mL) was mixed with one mL of 65 mmol L−1 sodium phosphate buffer (pH 7.8) and 0.2 mL of 10 mmol L−1 hydroxyl ammonium chloride. The resulting supernatant was incubated 25 °C for 20 min, and the above reaction mixture (one mL) was mixed with one mL of 4-aminobenzene sulfonic acid (17 mmol L−1) and one mL of α-naphthylamine (seven mmol L−1), then incubated at 30 °C for 30 min. The absorbance at 530 nm was measured with a 756C-UV-VIS spectrophotometer (Shanghai Spectrum Instruments Co. Ltd, Shanghai, China) (Elstner & Heupel, 1976).

Fresh foxtail millet leaves (0.1 g) were homogenized with five mL of chilled acetone in an ice bath and centrifuged at 4,000 rpm for 15 min. The supernatant (one mL) was mixed with 0.1 mL of 20% TiCl4 concentrated hydrochloric acid and 0.2 mL of concentrated ammonium hydroxide. The reaction mixture was centrifuged at 10,000 rpm for 10 min and the precipitate was dissolved in three mL H2SO4 (one mol L–1). The absorbance of the supernatant at 410 nm was monitored with a 756C-UV-VIS spectrophotometer (Zhang et al., 2012).

Determination of antioxidant enzyme activities

Fresh foxtail millet leaves (0.1 g) were homogenized with two mL of sodium phosphate buffer (pH 7.0, containing 0.1 mmol L−1 EDTA and 1% PVP (w/v)) in an ice bath. The homogenate was centrifuged at 10,000 rpm for 15 min at 4 °C. The supernatant was extracted and used to measure the activities of superoxide (SOD), peroxidase (POD), catalase (CAT), and ascorbate peroxidase (APX). The samples were measured with a 756C-UV-VIS spectrophotometer.

Superoxide activity was analyzed by using the nitro blue tetrazolium (NBT) method. The reaction mixture (five mL) consisted of phosphate buffer (50 mmol L−1, pH 7.8), L-methionine (13 mmol L−1), NBT (0.075 mmol L−1), EDTA (0.1 mmol L−1), riboflavin (0.002 mmol L−1), and 20 μL of enzyme extract. The reaction mixture was then illuminated under 4000 lx for 15 min at 25 °C. One unit of activity corresponds to the amount of protein required to inhibit 50% of the initial reduction of NBT under light conditions. The absorbance at 560 nm was measured, and a non-irradiated complete reaction mixture served as the control. For POD activity determination, 20 µL of enzyme extract was added into three mL of reaction liquid containing three mL of sodium phosphate buffer (100 mmol L−1, pH 6.0), 19 µL of guaiacol, and 28 µL of 30% H2O2. POD activity was measured by the changes in the absorbance of the reaction solution at 470 nm for 3 min. For CAT activity determination, the reaction liquid consisted of 2.7 mL of Tris-HCl (50 mmol L−1, pH 7.0), 50 µL of H2O2 (200 mmol L−1), and 20 µL of enzyme extract. Enzyme activity was continuously determined at 240 nm for 3 min following H2O2 decomposition. In accordance with the method of APX activity determination by Nakano & Asada (1981), the reaction mixture consisted of three mL sodium phosphate buffer (50 mmol L−1, pH 7.0), 0.4 mL of EDTA (0.3 mmol L−1), one mL of ascorbate (0.9 mmol L−1), and one mL of enzyme extract. The reaction mixture was then incubated for 5 min at 25 °C. The above reaction mixture was mixed with 0.5 mL of H2O2 (0.25 mmol L−1), and H2O2 decomposition was measured by the decline in absorbance at 290 nm for 1 min.

Glutathione reductase (GR) activity was analyzed according to Halliwell & Foyer (1978). Fresh foxtail millet leaves (0.1 g) were ground in an ice bath with two mL of Tris-HCl (pH 7.5, containing 1% PVP and 0.1 mmol L−1 EDTA), and centrifuged at 10,000 rpm for 15 min at 4 °C. The enzyme extract (150 μL) was added into three mL of reaction liquid including Tris-HCl (pH 7.5), MgCl2 (three mmol L−1), GSSG (0.5 mmol L−1), and NADPH (0.15 mmol L−1). Enzyme activity was calculated by the change in absorbance at 340 nm for 3 min.

Determination of photosynthetic gas exchange and chlorophyll

Photosynthetic rate (PN), transpiration rate (E), and stomatal conductance (Gs) were measured by CI-340 portable photosynthesis system (CID Bio-Science, Inc., Camas, WA, USA) from 9:30 to 10:30 am. The photosynthetically active radiation (PAR) at the leaf surface was approximately 1,000 ± 50 µmol/m2/s, the temperature of the leaf chamber was 30 ± 2 °C, and the ambient CO2 concentration was 380 ± 50 µmol/mol.

To measure photosynthetic pigments, fresh leaves (0.1 g) were soaked in 10 mL of ethanol (96%, v/v) and stored in the dark for 24 h. The absorbance of the supernatants was measured at 649 and 665 nm with a 756C-UV-VIS spectrophotometer (Lichtenthaler, 1987).

Measurement of chlorophyll fluorescence and P700 parameters

The chlorophyll fluorescence and P700 parameters were simultaneously detected using a luminoscope (Dual PAM-100, WALZ, Effeltrich, Germany) (Pfündel, Klughammer & Schreiber, 2008). After dark adaptation for 30 min, the kinetics of chlorophyll fluorescence induction and P700 oxidation were simultaneously detected according to the “Fluo + P700” analysis mode.

First, the minimal fluorescence (F0) was detected under weak light (seven µmol/m2/s). Subsequently, the maximum fluorescence (Fm) was determined by the saturation pulse (4,000 μmol/m2 /s, 800 ms) method. The slow induction curve was calculated as follows: Fv/Fm = (Fm − F0)/Fm, ETRII = PAR × 0.84 × 0.5 × YII, YII = (F′m – F)/F′m, YNO = 1/(NPQ + 1 + qL (Fm/F0 − 1)), YNPQ = 1 − YII − YNO, where Fv/Fm and YII are the maximum PSII quantum yield and effective PSII quantum yield, respectively, and YNPQ and YNO are the regulated energy dissipation and non-regulated energy dissipation, respectively.

P700 oxidation was monitored by the changes in the transmittance signals at 875 and 830 nm (Klughammer & Schreiber, 2008). The maximal P700 (Pm) was denoted by the maximal P700 signal reduction to full oxidation. YNA, the nonphotochemical quantum yield of photosystem I (PSI), is a measure of the acceptor-side limitation and calculated according to the formula YNA = (P′m − Pm)/P′m. YI, the photochemical quantum yield of PSI, was calculated according to the formula YI = (P′m− P)/Pm. YND, the nonphotochemical quantum yield of PSI, is a measure of the donor side limitation and calculated as YND = (P − P0)/Pm. The sum of these three quantum yields is one, i.e.: YI + YND + YNA = 1. The electron transfer efficiency of PSI, ETRI was assessed by the Dual PAM software.

Measurement of endogenous hormones

Fresh foxtail millet leaves (0.5 g) were homogenized with four mL of 80% methanol (one mmol/L 2, 6-di-tert-butyl-4-methylphenol) in an ice bath. The homogenate mixture was refrigerated at 4 °C for 4 h, and centrifuged at 3,500 rpm for 8 min. The precipitate was added to one mL of 80% methanol, refrigerated at 4 °C for 1 h, and centrifuged. The supernatant passed through a C18 solid phase extraction column, and then dried with nitrogen. The hormone extract was obtained by mixing with one mL of the sample diluent (PBS containing 0.1% Tween-20, gelatin one µg/L) and determined using enzyme-linked immunosorbent assay. The hormone assay kit was supplied by China Agricultural University, and the content of auxin (IAA), abscisic acid (ABA), zeatin (ZR), and gibberellin (GA) contents were measured by using the Thermo Multiskan FC enzyme-labeled instrument.

Statistical analysis

Statistical analyses were performed using Statistical Product and Service Solutions 19.0 (SPSS Inc., Chicago, IL, USA). Quantitative data were expressed as the mean ± standard error, and multiple comparisons were analyzed by Duncan’s multiple range test. P < 0.05 was considered significantly different. In this study, Fisher discriminant and principal component analyses were calibrated by using the selected indices with the highest difference between the different herbicide treatments and control, to simplify the indicators.

Results

Effects of fluroxypyr on growth parameters

After exposing the seedlings to 0.5–4.0 L ai ha−1 fluroxypyr for 5 days, the plant height of Zhangza 5 significantly differed between the control and herbicide treatments. At 10 days post-fluroxypyr treatment, the plant height increased by 10.16%, 12.62%, and 5.73% after exposure to 0.5, 1.0, and 2.0 L ai ha−1, respectively. As shown in Table 1, the leaf area of Zhangza 5 significantly differed between the recommended dose (one L ai ha−1) and other treatments after 5 days. A similar response of the leaf area was recorded when Zhangza 5 was exposed to fluroxypyr for 10 days, thereby showing significant difference among treatments compared with the recommended dosage.

Table 1 Effect of fluroxypyr application on growth parameters in foxtail millet.

Herbicide dosage (L ai ha−1)	Plant height (cm)	Leaf area (cm2)	
5 Day	10 Day	5 Day	10 Day	
0	26.05 ± 0.23c	30.50 ± 0.34c,d	8.00 ± 0.48b	12.10 ± 0.15b	
0.5	29.95 ± 0.46b	33.60 ± 0.57a,b	9.71 ± 0.07b	12.50 ± 0.18b	
1	32.50 ± 0.02a	34.35 ± 0.23a	13.94 ± 0.43a	15.49 ± 0.32a	
2	28.60 ± 1.41b	32.25 ± 0.95b,c	8.70 ± 0.69b	12.18 ± 0.36b	
4	23.60 ± 1.21d	30.10 ± 0.02d	7.78 ± 1.77b	10.75 ± 0.33c	
Note:

Different letters in the same column indicate a significantly difference at the P < 0.05 level by Duncan’s new multiple range test.

Effects of fluroxypyr on stress parameters

Fluroxypyr treatment at doses ranging from zero to four L ai ha−1 increased H2O2 and O2− accumulation in plants, with the maximum expression levels obtained after treatment with a dosage of four L ai ha−1 for 5 and 10 days. As shown in Table 2, maximum H2O2 and O2− accumulations significantly increased by treatment with four L ai ha−1 fluroxypyr for 5 and 10 days compared with the control. Similarly, treatment with fluroxypyr at the recommended dose (one L ai ha−1) increased H2O2 and O2− contents by 1.01- and 1.02- fold relative to the control after 5 days and by 2.02- and 1.05-fold relative to the control after 10 days, respectively. In our studies, we observed a time-dependent decrease in reactive oxygen species (ROS) in response to fluroxypyr exposure.

Table 2 Effect of fluroxypyr application on stress parameters in foxtail millet.

Herbicide dosage (L ai ha−1)	H2O2 content (µmol g−1 FM)	O2− generating rate (nmol g−1 min−1 FM)	
5 Day	10 Day	5 Day	10 Day	
0	40.70 ± 0.02c	15.53 ± 0.05e	164.18 ± 1.09d	153.06 ± 0.41c	
0.5	40.83 ± 0.02c	23.69 ± 0.03d	167.26 ± 0.34c	159.45 ± 0.95b	
1	41.16 ± 0.03c	31.40 ± 0.05c	168.20 ± 0.54c	160.87 ± 0.95b	
2	42.96 ± 0.01b	36.40 ± 0.06b	177.19 ± 0.85b	162.05 ± 0.54b	
4	51.30 ± 0.03a	39.06 ± 0.07a	193.28 ± 0.41a	166.07 ± 1.63a	
Notes:

Different letters in the same column indicate a significantly difference at the P < 0.05 level by Duncan’s new multiple range test.

FM, fresh mass.

Effects of fluroxypyr on enzyme activities

As shown in Fig. 1, the SOD, POD, CAT, APX, and GR activities significantly increased. The exposure of Zhangza 5 to increasing levels of fluroxypyr led to increases in SOD, CAT, APX, and GR activities. However, further increases in fluroxypyr concentration beyond two L ai ha−1 failed to promote SOD, CAT, APX, and GR accumulation. The activities of SOD, CAT, APX, and GR significantly increased after treatment with two L ai ha−1 fluroxypyr relative to the control. POD activities were stimulated after exposure to different fluroxypyr concentrations, and its effect was concentration-dependent; the highest POD activity was observed at four L ai ha−1 fluroxypyr. POD activities significantly increased by 2.91- and 3.53-fold relative to the control after treatment with four L ai ha−1 fluroxypyr for 5 and 10 days, respectively.

Figure 1 Effect of fluroxypyr on superoxide, peroxidase, catalase, ascorbate peroxidase, and glutathione reductase activities of foxtail millet.

(A) SOD activity of foxtail millet after 5 d or 10 d of fluroxypyr treatment; (B) POD activity of foxtail millet after 5 d or 10 d of fluroxypyr treatment; (C) CAT activity of foxtail millet after 5 d or 10 d of fluroxypyr treatment; (D) APX activity of foxtail millet after 5 d or 10 d of fluroxypyr treatment; (E) GR activity of foxtail millet after 5 d or 10 d of fluroxypyr treatment. SOD, POD, CAT, APX, and GR represent superoxide, peroxidase, catalase, ascorbate peroxidase, and glutathione reductase activities, respectively.

Effects of fluroxypyr on net photosynthetic rate and pigments

As shown in Table 3, PN, E, Gs, and Chl showed a progressive decrease with increasing fluroxypyr concentration. After one L ai ha−1 (recommended dose) fluroxypyr treatment for 5 days, PN and E significantly decreased by 23.72% and 22.56% compared with the control, respectively. However, one L ai ha−1 fluroxypyr treatment for 5 days inhibited Gs by 3.53%. Moreover, after with one L ai ha−1 fluroxypyr for 10 days, PN, E, and Gs of the treated plants were reduced by 18.78%, 15.12%, and 18.79% compared with the control, respectively. According to our results, significant differences were recorded between the treatments and control. Chlorophyll content was also significantly affected following fluroxypyr treatment; however, no significant differences were observed between the control and treatment using the recommended dose after 5 and 10 days of exposure.

Table 3 Effect of fluroxypyr application on photosynthetic parameter and pigments in leaves of foxtail millet.

Herbicide dosage
(L ai ha−1)	PN (µmol(CO2)m−2s−1)	E (µmol(H2O)m−2s−1)	Gs (µmol(CO2)m−2)	Chl (mg g−1)	
5 Day	10 Day	5 Day	10 Day	5 Day	10 Day	5 Day	10 Day	
0	16.86 ± 0.31a	26.56 ± 0.14a	2.57 ± 0.01a	4.76 ± 0.09a	67.53 ± 0.18a	120.99 ± 2.76a	9.57 ± 1.13a	14.24 ± 0.75a	
0.5	14.81 ± 0.52b	23.81 ± 0.73b	2.17 ± 0.06a,b	4.31 ± 0.02b	61.81 ± 2.09b	108.81 ± 0.95b	9.26 ± 0.03a	13.82 ± 0.24a	
1	12.86 ± 0.48c	21.57 ± 0.91b	1.99 ± 0.32b	4.04 ± 0.01c	65.14 ± 0.79a,b	98.25 ± 2.63c	8.33 ± 0.50a,b	13.55 ± 0.09a	
2	11.88 ± 0.47c,d	22.27 ± 1.13b	1.72 ± 0.09b,c	3.85 ± 0.10c	51.67 ± 1.93c	91.68 ± 4.74c	7.17 ± 0.07b,c	11.75 ± 0.24b	
4	11.28 ± 0.47d	16.95 ± 0.36c	1.24 ± 0.04c	3.16 ± 0.03d	50.03 ± 1.63c	74.71 ± 2.55d	5.83 ± 0.21c	10.92 ± 0.42b	
Note:

Different letters in the same column indicate a significantly difference at the P < 0.05 level by Duncan’s new multiple range test. PN, E, Gs and Chl represent photosynthetic rate, Transpiration rate, stomatal conductance, chorophyll content, respectively.

Effects of fluroxypyr on chlorophyll fluorescence and P700 parameters

After 5 days of exposure to fluroxypyr, the Fv/Fm of treated plants decreased to 0.13%, 0.32%, 1.71%, and 3.29% compared with the control, and the differences between treatments were insignificant. Moreover, after 10 days, the Fv/Fm of the treated plants recovered to control levels. As shown in Fig. 2, the changes in Y(II) and ETR(II) were consistent under fluroxypyr treatment, and the values were reduced in a dose-dependent manner. After 5 and 10 days of treatment, Y(II) decreased from 12.44% to 22.96%, and from 18.25% to 32.14%, respectively, and ETR(II) decreased from 12.37% to 22.69%, and from 14.91% to 32.31%, respectively. Additionally, Y(NPQ) increased with increasing fluroxypyr doses, and the maximum accumulation was observed at two L ai ha−1. During treatment with two L ai ha−1 fluroxypyr, the Y(NPQ) values were 36.39%, and 50.75%, compared with the control.

Figure 2 Effect of fluroxypyr on chlorophyll fluorescence and P700 parameters of foxtail millet.

(A) Fv/Fm of foxtail millet after 5 d or 10 d of fluroxypyr treatment; (B) ETR(II) and Y(II) of foxtail millet after 5 d or 10 d of fluroxypyr treatment; (C) Y(NPQ) and Y(NO) of foxtail millet after 5 d or 10 d of fluroxypyr treatment; (D) Pm of foxtail millet after 5 d or 10 d of fluroxypyr treatment; (E) ETR(I) and Y(I) of foxtail millet after 5 d or 10 d of fluroxypyr treatment; (F) Y(ND) and Y(NA) of foxtail millet after 5 d or 10 d of fluroxypyr treatment. Fv/Fm, maximum quantum yield of PS; Y(II), PSII effective quantum yield; ETR(II), PSII electron transport rate. Y(NO), quantum yield of nonregulated energy dissipation in PSII; Y(NPQ), quantum yield of regulated energy dissipation in PSII. Pm, maximal P700 change; Y(I), photochemical quantum yield of PSI; ETR(I), PSI electron transport rate; Y(NA), quantum yield of nonphotochemical energy dissipation due to acceptor-side limitation in PSI; Y(ND), quantum yield of nonphotochemical energy dissipation due to donor-side limitation in PSI.

PSI activities were substantially affected by fluroxypyr treatment. As shown in Fig. 2, Pm, Y(I), and ETR(I) were evidently decreased with increasing fluroxypyr dosage; however, no significant differences between the control and herbicide-treated plants were observed after 5 days of exposure. Treatment with increasing fluroxypyr concentration increased the Y(ND) of Zhangza 5. The maximum Y(ND) level was observed at fluroxypyr treatment. However, further increasing the dosages of fluroxypyr beyond two L ai ha−1 no longer increased Y(ND).

Effects of fluroxypyr on endogenous hormones

Fluroxypyr treatment at doses ranging from 0.5 to 4 L ai ha−1 increased IAA and ABA accumulation in plants, with the maximum accumulation observed at four L ai ha−1 (Fig. 3). The application of four L ai ha−1 fluroxypyr significantly increased IAA and ABA by 26.69% and 23.36% relative to that of the control after 5 days of treatment and 5.37% and 35.53% relative to that of the control after 10 days of treatment, respectively. A similar response was observed between GA and ZR accumulation in Zhangza 5 exposure to fluroxypyr; GA and ZR decreased with increasing fluroxypyr concentration, and the difference observed between the control and treated plants was significant. After treatment with one L ai ha−1 (recommended dose) fluroxypyr for 5 and 10 days, the GA and ZR values significantly decreased by 10.46%, 18.72%, and by 13.50%, 23.59% relatively to the control treatment, respectively.

Figure 3 Effect of fluroxypyr on endogenous hormones of foxtail millet.

(A) IAA of foxtail millet after 5 d or 10 d of fluroxypyr treatment; (B) GA of foxtail millet after 5 d or 10 d of fluroxypyr treatment; (C) ABA of foxtail millet after 5 d or 10 d of fluroxypyr treatment; (D) ZR of foxtail millet after 5 d or 10 d of fluroxypyr treatment. IAA, GA, ABA, and ZR represent auxin, gibberellin, zeatin, abscisic acid, respectively.

Comprehensive evaluation of fluroxypyr on various foxtail millet indices

To further comprehensively evaluate the effects of fluroxypyr on various foxtail millet indices, Fisher discriminant analysis was used. Herbicide treatment was used as the grouping variable Y, and the physiological parameters were used as the independent variable X.

(1) Fisher discrimination was used to evaluate the seven indicators of foxtail millet leaf, namely, H2O2(X1), O2− (X2), SOD(X3), POD(X4), CAT(X5), APX(X6), and GR(X7). The following discriminant functions were obtained: Y1 = −3.850 + 4.868X3 + 3.588X6 − 48.734X7

Y2 = −0.344 − 1.408X3 + 2.886X6 − 7.736X7

Y3 = −5.062 − 0.107X3 + 0.020X6 + 12.487X7

The cumulative contribution rate of the first-class functions, which is the main function, reached 98.8%. Thus, SOD(X3), POD(X4), and GR(X7) were used as indices to evaluate the peroxidation characteristics of foxtail millet.

(2) Fisher discrimination was used to evaluate the fourteen indicators of foxtail millet leaf, namely, PN(X1), E(X2), Gs(X3), Chl(X4), Fv/Fm(X5), Y(II)(X6), ETR(I)(X7), Y(NPQ)(X8), Y(NO)(X9), Pm(X10), Y(I)(X11), ETR(I)(X12), Y(ND)(X13), and Y(NA)(X14). The following discriminant functions were obtained: Y1 = −252.994 + 427.640X6 − 66.364X8 + 143.154X10 + 1.396X12

Y2 = −69.158 + 151.904X6 + 96.548X8 + 22.743X10 − 0.036X12

Y3 = −5.251 − 91.400X6 − 1.385X8 + 7.042X10 + 0.184X12

Y4 = 1.257 + 2.323X6 − 1.378X8 + 3.767X10 − 0.018X12

The cumulative contribution rate of the first-class functions, which is the main function, reached 94.4%. Thus, Y(II)(X6), Y(NPQ)(X8), Pm(X10), and ETR(I)(X12) were used as indices to evaluate the photosynthetic characteristics of foxtail millet.

(3) Fisher discrimination was used to evaluate the four indicators of foxtail millet leaf, namely, IAA(X1), GA(X2), ZR(X3), and ABA(X4). The following discriminant functions were obtained: Y1 = −78.315 + 8.217X2 + 0.229X4

Y2 = −13.841 + 1.139X2 + 0.073X4

The cumulative contribution rate of the first-class functions, which is the main function, reached 94.4%. Thus, GA(X2) and ABA(X4) were used as indices to evaluate the endogenous hormone characteristics of foxtail millet.

As shown in Table 4, the cumulative contribution rate of the first two principal components reached 93.72%. Thus, the first two principal components were further analyzed. The first principal component accounted for 59.23% of the variance observed, whereas the second principal component accounted for 34.49% of this variance. The load values of ERT (I), Y(II), and GA, which were 0.978, 0.975 and 0.963, respectively, were the largest in the first principal component. ABA also showed high absolute loading value in the first principal component, but exhibited a negative effect. Therefore, the first principal component could explain the photosynthetic characteristics and endogenous hormones of foxtail millet. In the second principal component, the loading values of plant height and leaf area were relatively high and could represent the growth parameters of foxtail millet.

Table 4 Component matrix and variance explained by principal component.

Traits	Component weight	
Component 1	Component 2	
Plant height	0.189	0.975	
Leaf area	0.146	0.888	
SOD activity	0.175	0.965	
POD activity	−0.926	−0.245	
GR activity	−0.702	0.640	
Y(II)	0.963	−0.233	
Y(NPQ)	−0.652	0.653	
Pm	0.944	0.255	
ETR(I)	0.978	−0.206	
GA	0.975	0.184	
ABA	−0.963	−0.178	
Eigenvalue	6.515	3.794	
Cumulative contribution (%)	59.230	93.722	

According to the principal component analysis, the model was set up as follows: F = 0.592Z1 + 0.345Z2. The comprehensive evaluation scores of the two and four L ai ha−1 fluroxypyr treatments were relatively low, which indicated that high fluroxypyr concentrations negatively affected the photosynthetic characteristics and endogenous hormones of foxtail millet. Furthermore, the comprehensive evaluation scores of the 0.5 and 1 L ai ha−1 fluroxypyr treatments were 0.941 and 2.007, respectively, and the scores of first two principal components were positive, which indicated that the moderate concentrations of fluroxypyr positively affected the peroxidation characteristics, photosynthetic characteristics, and endogenous hormones of foxtail millet (Table 5). Consequently, the scores of the two factors in one L ai ha−1 fluroxypyr treatments were both relatively high and represented the comprehensive effect of herbicide in foxtail millet (Fig. 4).

Figure 4 PCA score plot, the percentages of PC1 and PC2 representing the total variances of samples.

((1)–(3)) represent fluroxypyr dosage four L ai ha−1, ((4)–(6)) represent fluroxypyr dosage two L ai ha−1, ((7)–(9)) represent 0.5 L ai ha−1, ((10)–(12)) represent fluroxypyr dosage one L ai ha−4, and ((13)–(14)) represent fluroxypyr dosage zero L ai ha−1, respectively.

Table 5 Principal score and general score of comprehensive evaluation.

Treatment	Prin1 score	Prin2 score	Composite score F	
0.5	0.083	2.585	0.941	
1	2.998	0.672	2.007	
2	−4.242	0.638	−2.291	
4	−7.850	−2.110	−5.375	

Discussion

Numerous studies have demonstrated that excessive application of fluroxypyr in the environment inhibits plant growth (Wu et al., 2010; Guo et al., 2018). In the present study, we found that fluroxypyr had a slight effect on plant growth at relatively low doses, but exerted significant influence at high doses. The growth inhibition of fluroxypyr is closely related to the accumulation of O2− and H2O2 in plants (Wu et al., 2010). According to the inhibitory effect of fluroxypyr on ROS, the present study showed that the maximum accumulation of H2O2 and O2− occurs when plants are treated with four L ai ha−1 fluroxypyr for 5 and 10 days.

Herbicide-induced oxidative stress causes cellular peroxidation and molecular damage through ROS over-accumulation (Kehrer, 1993; Chen et al., 2009). To inhibit ROS action, plants have evolved protective enzymatic mechanisms, such as those involving SOD, POD and CAT (Mahaboob Khan & Kour, 2007). In most cases, the enzyme-based antioxidant system is one of the important ways for plants to resist environmental stress, which reflects not only the level of toxicity but also the ability to tolerate stress. Our analyses showed that the SOD, POD, CAT, APX, and GR activities generally increased at low fluroxypyr dosages but decreased at high fluroxypyr dosages, thereby reflecting an increased degree of oxidative stress.

Several symptoms related to changes in plant photosynthesis were observed in this work. After fluroxypyr treatment, one such visible symptom was chlorosis, which showed that chlorophyll was certainly sensitive to fluroxypyr exposure. In the present study, chlorophyll formation was considerably suppressed at high concentrations (two and four L ai ha−1). A reduction in PN, as reflected by the decrease in E and Gs, was also noted after fluroxypyr treatment. These results suggested that fluroxypyr destroyed the chloroplast structure of foxtail millet, increases the risk of photo-oxidation damage, and reduced light absorption, transmission, and distribution between PSII and PSI (Havaux, Strasser & Greppin, 1991).

Chlorophyll and gas exchange measurements are nondestructive methods used to identify the modes of action of certain herbicides in the ecological risk assessment of herbicide exposure. In this study, chlorophyll fluorescence showed that fluroxypyr treatment affected photosynthesis efficiency, even in the absence of phenotypic symptoms (Qian et al., 2014). Our analyses showed that each fluroxypyr treatment caused different degrees of Fv/Fm decrease, and the differences among treatments after 5 days of exposure were insignificant. Compared with the control, only the maximum fluroxypyr dose (fourfold the recommended dose) showed significantly reduced Fv/Fm values. Given that fluroxypyr inhibited electron transport in Zhangza 5, it also caused the decline in Y(II) and ETR (II). Y(NPQ) represents the regulated photo-protective NPQ mechanism, namely, dissipating the fraction of energy in the form of heat (Christof & Ulrich, 2008). Y(NPQ) also dissipates excess PSII energy via the xanthophyll cycle and associated carotenoids (Deng et al., 2013). In the present study, fluroxypyr treatment caused the increase in Y(NO), which demonstrated that the plant cannot protect itself against damage from excess illumination due to blocking the regulatory mechanism of the non-photochemical dissipation of energy. Consistent with our results, Guo et al. (2018) reported that sethoxydim has similar effects on the photosynthetic pigments, photosynthetic gas exchange and chlorophyll fluorescence parameters of foxtail millet.

Previous studies on several plant species have demonstrated that a typical feature of PSI photoinhibition is the reduction in the maximum oxidation reduction ability of PSI (Scheller & Haldrup, 2005). In the present study, Pm, Y(I), and ETR(II) decreased with increasing fluroxypyr doses, and were significantly inhibited compared with those of the control. Fluroxypyr inhibited electron transport in Zhangza 5, which caused an increase in Y(NA). Y(NA) is an important photo-damage indicator in PSI, and it is affected by dark adaptation and level of damage to CO2 fixation. Y(NA) was significantly increased by treatment with >2 L ai ha−1 fluroxypyr dose, showing that fluroxypyr aggravated the injury of PSII in foxtail millet leaves, reduced electronic accumulation of acceptor-side in PSI, blocked the dark reaction process, and decreased the fixed amount of CO2. Y(ND) reflected the state of electron donors in PSI and is affected by the transmembrane proton gradient and degree of damage of PSII (Yuan et al., 2013). In our study, Y(ND) in Zhangza 5 increased at the fluroxypyr concentrations of ≤1 L ai ha−1.

To further understand the mechanism of fluroxypyr toxicity about Zhangza 5, we analyzed the effect of fluroxypyr on endogenous hormones. Herbicides induced changes in endogenous hormones and regulated stomatal behavior. Zhou et al. (2014) reported that the stomatal behavior of plant leaves is related not only to their content of endogenous hormones, but also to the balance of various hormones. Wang, Zhou & Zhou (1994) reported that stomatal closure and transpiration reduction are affected by ABA, and also result from the combined action of ABA and ZR. In the present study, the IAA and ABA contents in the plant leaves were significantly increased; however, the GA and ZR contents significantly decreased. In our study, Gs values showed a progressive decrease with the increase in fluroxypyr concentrations, while ABA content gradually accumulated with the increase in fluroxypyr concentrations, implying that ABA inhibited photosynthesis by decreasing stomatal conductance. In addition to stomatal regulation, ABA and ZR also directly influenced photosynthesis. ABA reduced the electric potential of the chloroplast cell membrane, thereby reducing photosynthetic electron transport, whereas ZR could increase the Rubisco activity and promote photosynthetic electron transport.

High efficiency and safety are the main objectives of herbicide application. A total of 11 simplified indicators were screened by Fisher discriminant analysis, and principal component analysis was used to comprehensively evaluate the growth parameters, peroxidation characteristics, photosynthetic characteristics and endogenous hormones of foxtail millet at different fluroxypyr dosages. The model was set up as follows: F = 0.592Z1 + 0.345Z2. The cumulative contribution rate of the first two principal components reached 93.72%. The first principal component explained 59.23% of the variance including photosynthetic characteristics and endogenous hormones of foxtail millet. The second principal component accounted for 34.49% of the variance and had high loadings for growth parameters. Consequently, these two principal components were selected to represent the comprehensive effect of fluroxypyr on foxtail millet.

Conclusions

The effect of fluroxypyr on foxtail millet is complex and involves combination of several factors. The application of one L ai ha−1 fluroxypyr exerted minimal effects on growth parameters, oxidase activity, photosynthetic activity, and endogenous hormones, indicating its efficient and safe benefits in foxtail millet application.

Supplemental Information

Supplemental Information 1 The raw data has been supplied as Supplemental Dataset Files.

Raw data refer to the results of fluroxypyr applied for data analyses and preparation for the detailed investigation shown Tables 1–3, Figs.1–3 for the time period of 5 and 10 days.

Click here for additional data file.

We are gratefully acknowledgment help from Professor Yuguo Wang, Shanxi Agricultural University for suggestions on the manuscript. Thanks are also due to three anonymous reviewers and editor for their thoughtful and valuable comments and suggestions, which helped in improving the manuscript.

Additional Information and Declarations

Competing Interests

Author Contributions

Data Availability

The authors declare no conflict of interest.

Meijun Guo performed the experiments, analyzed the data, prepared figures and/or tables, authored or reviewed drafts of the paper, approved the final draft.

Jie Shen contributed reagents/materials/analysis tools, authored or reviewed drafts of the paper, approved the final draft.

Xi-e Song contributed reagents/materials/analysis tools, authored or reviewed drafts of the paper, approved the final draft.

Shuqi Dong contributed reagents/materials/analysis tools, authored or reviewed drafts of the paper, approved the final draft.

Yinyuan Wen contributed reagents/materials/analysis tools, authored or reviewed drafts of the paper, approved the final draft.

Xiangyang Yuan conceived and designed the experiments, authored or reviewed drafts of the paper, approved the final draft.

Pingyi Guo conceived and designed the experiments, authored or reviewed drafts of the paper, approved the final draft.

The following information was supplied regarding data availability:

Raw data is available in the Supplemental Files.

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
