# Peer review of "Comprehensive evaluation of fluroxypyr herbicide on physiological parameters of spring hybrid millet"

_PeerJ, doi:10.7717/peerj.7794_

## Round 0.1 · original submission · Major Revisions

Dear author,

Your paper has been assessed by three reviewers and myself as academic Editor.

As you could see below, the manuscript needs a major revision.
Please address all concerns of the three reviewers and submit a revised version of the manuscript. Please include a detailed response to each reviewer.

English language should be corrected by a native speaker with scientific expertise.

[]

Reviewer 1 ·

Basic reporting

The paper entitled “Comprehensive evaluation of fluroxypyr herbicide on physiological parametric in spring hybrid millet” describe the physiological and biochemical changes occurred in Setaria italica by the application of a hormonal herbicide (fluroxypyr). The importance of this paper designs a technical guide to the use of chemical control for this crop.

For this objective, the authors developed a very detailed study to measure these changes after the herbicide application (5 and 10 d). However, the introduction has critical and conceptual errors. The principal argument is the study of the resistance mechanisms in Foxtail millet. However, the resistance is caused by the repeated application of herbicides; in other words, is the result of a selection process. In this case, the term tolerance is adequate for the study. The Weed Science Society of America defines tolerance as “the inherent ability of a species to survive and reproduce after herbicide treatment. This implies that there was no selection or genetic manipulation to make the plant tolerant; it is naturally tolerant (Weed Technology 12(4):789, 1998). For this reason, the introduction and objectives are not congruent.
If authors can change the scope of the paper, this will be an important contribution to weed management.

Experimental design

the experimental design is adequate for evaluating the "tolerance" of Foxtail millet to the herbicide fluroxypyr. Several physiological parameters were evaluated and give a detailed description of changes occurs in plants.

Validity of the findings

'no comment'

Additional comments

This is a good work. However, the introduction has mistakes in the concepts. If you can make the changes, this paper could be accepted whatever journal.

Reviewer 2 ·

Basic reporting

In general, this work appears to be well conducted and presenter. However, my main criticism is the excesive wording in introduction and discussion.

Experimental design

Plase explain why is the recommeded dose is of 1 L/ha doses up to 4 l/Ha were used. Severe damage are expected under such treatments.

Validity of the findings

No comments. Just please, try to present them ia a succint way.

Additional comments

The work deserve publication, but it could be improved by shortening as suggested.

Reviewer 3 ·

Basic reporting

The manuscript (ID: 34816) entitled " Comprehensive evaluation of fluroxypyr herbicide on physiological parametric in spring hybrid millet " is of high quality and suitable for publication in this journal, however it needs minor revisions.
This manuscript describes a study on the physiological reaction of different physiological parametric to the spraying of different dosages of herbicide fluroxypyr at the seedling stage of hybrid foxtail millet. Effects of an auxin herbicide, fluroxypyr, on growth parameters, photosynthetic status, and hormone status were examined. Among them, photosynthesis and hormone status (GA and ABA) were most sensitive. I evaluated the manuscript in two parts:
(1) The language is not checked properly, there are frequent grammatical errors and language editing by special editor of native speaker.
(2) The design, implementation and results: The manuscript has innovation but I do not understand what is the biochemical interpretation of the results of the statistical analyses. To be honest, I do not know if these methods are actually useful to address the resistance mechanism of this plant.

Experimental design

The authors treated foxtail millet with an herbicide fluroxypyr at various concentrations, and analyzed multiple aspects of the plant physiology. The plant not only showed enhanced and suppressed growth depending on the treated concentration, but also accumulated the reactive oxygen species, and showed enhanced enzyme activities that involved in the metabolism of reactive oxygene species. The treatment also affected various parameters concerning the photosynthesis and plant hormone levels. The design and results are feasible and reliable, which provide useful data for herbicide related research in foxtail millet and other crops.

Validity of the findings

It is well known that photosynthesis is very sensitive to any biotic and abiotic disturbances. In fact, photosynthesis is significantly reduced after treatment of any herbicides irrespective of their modes of action. Furthermore, as fluroxypyr is a typical auxin herbicide, increased ethylene biosynthesis and ABA accumulation shall be observed after fluroxypyr treatment. In general, auxin herbicides including fluroxypyr disturb auxin homeostasis and promote ethylene biosynthesis and accumulation of ABA, resulting in growth retardation and plant death. The design and result are generally fine, which provide useful data for herbicide related research in foxtail millet and other crops.

Additional comments

1. The discussion is not sufficient and a bit rigid in terms of its structure.
2. Format: some refs are wrong in the names of Chinese authors, given names were used as surnames.
3. In line 71, whether the experiment was carried out in a laboratory or greenhouse?
4. The figures are not well organized. For example, in Fig. 1 the SOD activity is expressed as line graph, but the POD activity is expressed as bar graph. This is very strange because they are both enzyme activities. All of the figures should be modified in an appropriate way.
5. There is a grammatical error in the manuscript. Please check it by a native English speaker.

---

## Round 0.2 · Minor Revisions

Dear author

I can read that you have addressed all the reviewers concerns. The reviewers comments have been responded adequately.

Nevertheless, the Section Editor had some concerns that need to be fixed prior acceptance. We strongly recommend a professional English language editing service.

Specifically, the Section Editor Julin Maloof said:

"There are multiple issues with the manuscript and it is not suitable for publication in its current form. There are numerous problems with English language which make the manuscript difficult to understand in places. Some of these are noted below but this is not a comprehensive list.

• The abstract needs work. The abstract should be understandable by non-experts, but parts of this one are too technical, for example the equation is not explained: what is F? And I assume that Z1 and Z2 represent the two principal components. Probably the equation should be removed and replaced with words for the abstract. Sometimes "comprehensive score" is used in the abstract and other times "comprehensive evaluation";I think these are referring to the same thing but the inconsistency makes it hard to understand.
• All abbreviations need to be defined, including GA, ABA, ZR, ai, etc, in main text, not just in legends.
• The abstract and conclusion mention "ecological benefits" but this manuscript has nothing to do with ecology.
• line 64 "Thus, we provide convincing evidence of the underlying mechanism of NTS resistance to fluroxypyr exposure..." NO, here we have correlations, not causation. There is nothing in the manuscript that shows underlying mechanisms.
• The experimental design is not clear. How many replicates? Was the treatment applied in a randomized design? Blocked design? etc...
• It is stated that "SPASS" software was used for statistical analysis. Do the authors mean "SPSS"?
• In the text description of the linear discriminant analysis it needs to be made clear what "Y" is. I assume plants are categorized based on herbicide dose, and Y indexes that. But this needs to be made clear.
• line 282 " reflecting toxic herbicide enzymes" reword; current wording makes it sound as if herbicides have enzymes.
• line 291 "characteristics of internality." reword, this is not good English.
• line 292 "even in phenotypic symptom defect" reword, this does not make sense.
• line 308 "After dark adaption, the Calvin-Benson circle was damaged due to the inactivation of the key enzymes and electronic accumulation of acceptor-side in PSI, which caused the increase in Y (NA)." The manuscript does not show this. If this is just a general description of what Y(NA) indicates that needs to be made more clear.
• line 315 "changed" -> "changes"
• line 322 "which explains ABA content increases can lead to a decrease in stomatal conductance." Poorly worded.
• "RuBPcase" Rubisco is the more common abbreviation. Please consider changing.
• To better understand the principal components analysis please include a scatter plot, each plant represented by a point, colors or symbols indicate treatment does, x axis is PC1 and y axis is PC2
• lines 332- the discussion of the PCs is poorly worded. For example "the second principal component accounted for 34.49% of the variance contributed by its growth parameters." No! The PC does not account for 34.49% of the growth parameters. PC2 accounts for 34.49% of the overall variance in the data set."

Please correct those issues and be aware that the list from Dr Maloof is not comprehensive.

Reviewer 1 ·

Basic reporting

This article had been revised previously. The conceptual errors pointed out by this reviewer have been corrected.
In my point of view, I consider that this article has already the required quality to be published in PeerJ

Experimental design

no comment

Validity of the findings

no comment

Reviewer 2 ·

Basic reporting

All of my comments were properly adressed in this new version. In fact, I consider that the new version shows a significant improvement from the previous one, after the reviewer comments.

Experimental design

No comments

Validity of the findings

No comments

Additional comments

Addressing the reviewers' comments resulted in a better organized and clearer paper.

---

## Round 0.3 · Minor Revisions

Dear author
I can read that you have addressed the editorial concerns. His comments have been responded to adequately. Professional english editing has been performed.
Nevertheless the section editor Julin commented:

“This is very close and I appreciate the attention paid to my comments. However, for the new figure 4, instead of plotting the means of the plants in teach treatment it would be helpful to show either a point for each plant measured, or the standard error of the mean. We need this information to visualize how well separated the groups are based on the two PCs”

Please perform the required changes and submit a revised version.
I congratulate you for the nice piece of work, which will add value to PeerJ.

---

## Round 0.4 · accepted · Accept

Dear author

I can read that you have addressed the editorial concern. Figure 4 has been replaced as requested. During the production step please correct some language issues. There are still several sentences that could be improved for English style and meaning. For example, the last sentence of the abstract did not make sense for me: " had highest value of comprehensive evaluation, which had efficient and safe benefits in foxtail millet field."

Please check the whole manuscript while in production, with a native English speaker.